# Multilingual Neural Machine Translation with Knowledge Distillation

**Xu Tan**[1]*, **Yi Ren**[2]*, **Di He**[3], **Tao Qin**[1], **Zhou Zhao**[2] **& Tie-Yan Liu**[1]

[1]Microsoft Research Asia
{xuta,taoqin,tyliu}@microsoft.com

[2]Zhejiang University
rayeren,zhaozhou@zju.edu.cn

[3]Key Laboratory of Machine Perception, MOE, School of EECS, Peking University
di_he@pku.edu.cn

## Abstract

Multilingual machine translation, which translates multiple languages with a single model, has attracted much attention due to its efficiency of offline training and online serving. However, traditional multilingual translation usually yields inferior accuracy compared with the counterpart using individual models for each language pair, due to language diversity and model capacity limitations. In this paper, we propose a distillation-based approach to boost the accuracy of multilingual machine translation. Specifically, individual models are first trained and regarded as teachers, and then the multilingual model is trained to fit the training data and match the outputs of individual models simultaneously through knowledge distillation. Experiments on IWSLT, WMT and Ted talk translation datasets demonstrate the effectiveness of our method. Particularly, we show that one model is enough to handle multiple languages (up to 44 languages in our experiment), with comparable or even better accuracy than individual models.

## 1 Introduction

Neural Machine Translation (NMT) has witnessed rapid development in recent years (Bahdanau et al., 2015; Luong et al., 2015b; Wu et al., 2016; Gehring et al., 2017; Vaswani et al., 2017; Wu et al., 2018; Song et al., 2018; Shen et al., 2018; Guo et al., 2018; He et al., 2018; Gong et al., 2018), including advanced model structures (Gehring et al., 2017; Vaswani et al., 2017) and human parity achievements (Hassan et al., 2018). While conventional NMT can well handle single pair translation, training a separate model for each language pair is resource consuming, considering there are thousands of languages in the world[1]. Therefore, multilingual NMT (Johnson et al., 2017; Firat et al., 2016; Ha et al., 2016; Lu et al., 2018) is developed which handles multiple language pairs in one model, greatly reducing the offline training and online serving cost.

Previous works on multilingual NMT mainly focus on model architecture design through parameter sharing, e.g., sharing encoder, decoder or attention module (Firat et al., 2016; Lu et al., 2018) or sharing the entire models (Johnson et al., 2017; Ha et al., 2016). They achieve comparable accuracy with individual models (each language pair with a separate model) when the languages are similar to each other and the number of language pairs is small (e.g., two or three). However, when handling more language pairs (dozens or even hundreds), the translation accuracy of multilingual model is usually inferior to individual models, due to language diversity.

It is challenging to train a multilingual translation model supporting dozens of language pairs while achieving comparable accuracy as individual models. Observing that individual models are usually

---

*Authors contribute equally to this work.
[1]https://www.ethnologue.com/browse

of higher accuracy than the multilingual model in conventional model training, we propose to transfer the knowledge from individual models to the multilingual model with *knowledge distillation*, which has been studied for model compression and knowledge transfer and well matches our setting of multilingual translation. It usually starts by training a big/deep teacher model (or ensemble of multiple models), and then train a small/shallow student model to *mimic* the behaviors of the teacher model, such as its hidden representation (Yim et al., 2017; Romero et al., 2014), its output probabilities (Hinton et al., 2015; Freitag et al., 2017) or directly training on the sentences generated by the teacher model in neural machine translation (Kim & Rush, 2016a). The student model can (nearly) match the accuracy of the cumbersome teacher model (or the ensemble of multiple models) with knowledge distillation.

In this paper, we propose a new method based on knowledge distillation for multilingual translation to eliminate the accuracy gap between the multilingual model and individual models. In our method, multiple individual models serve as teachers, each handling a separate language pair, while the student handles all the language pairs in a single model, which is different from the conventional knowledge distillation where the teacher and student models usually handle the same task. We first train the individual models for each translation pair and then we train the multilingual model by matching with the outputs of all the individual models and the ground-truth translation simultaneously. After some iterations of training, the multilingual model may get higher translation accuracy than the individual models on some language pairs. Then we remove the distillation loss and keep training the multilingual model on these languages pairs with the original log-likelihood loss of the ground-truth translation.

We conduct experiments on three translation datasets: IWSLT with 12 language pairs, WMT with 6 language pairs and Ted talk with 44 language pairs. Our proposed method boosts the translation accuracy of the baseline multilingual model and achieve similar (or even better) accuracy as individual models for most language pairs. Specifically, the multilingual model with only $1/44$ parameters can match or surpass the accuracy of individual models on the Ted talk datasets.

## 2 BACKGROUND

### 2.1 NEURAL MACHINE TRANSLATION

Given a set of bilingual sentence pairs $D = \{(x, y) \in \mathcal{X} \times \mathcal{Y}\}$, an NMT model learns the parameter $\theta$ by minimizing the negative log-likelihood $-\sum_{(x,y) \in D} \log P(y|x; \theta)$. $P(y|x; \theta)$ is calculated based on the chain rule $\prod_{t=1}^{T_y} P(y_t|y_{<t}, x; \theta)$, where $y_{<t}$ represents the tokens preceding position $t$, and $T_y$ is the length of sentence $y$.

The encoder-decoder framework (Bahdanau et al., 2015; Luong et al., 2015b; Sutskever et al., 2014; Wu et al., 2016; Gehring et al., 2017; Vaswani et al., 2017) is usually adopted to model the conditional probability $P(y|x; \theta)$, where the encoder maps the input to a set of hidden representations $h$ and the decoder generates each target token $y_t$ using the previous generated tokens $y_{<t}$ as well as the representations $h$.

### 2.2 MULTILINGUAL NMT

NMT has been extended from the translation of a single language pair to multilingual translation (Dong et al., 2015; Luong et al., 2015a; Firat et al., 2016; Lu et al., 2018; Johnson et al., 2017; Ha et al., 2016), considering the large amount of languages pairs in the world. Some of these works focus on how to share the components of the NMT model among multiple language pairs. Dong et al. (2015) use a shared encoder but different decoders to translate the same source language to multiple target languages. Luong et al. (2015a) use the combination of multiple encoders and decoders, with one encoder for each source language and one decoder for each target language respectively, to translate multiple source languages to multiple target languages. Firat et al. (2016) share the attention mechanism but use different encoders and decoders for multilingual translation. Similarly, Lu et al. (2018) design the neural interlingua, which is an attentional LSTM encoder to bridge multiple encoders and decoders for different language pairs. In Johnson et al. (2017) and Ha et al. (2016), multiple source and target languages are handled with a universal model (one encoder and decoder), with a special tag in the encoder to determine which target language to translate. In Gu

et al. (2018a;b) and Neubig & Hu (2018), multilingual translation is leveraged to boost the accuracy of low-resource language pairs with better model structure or training mechanism.

It is observed that when there are dozens of language pairs, multilingual NMT usually achieves inferior accuracy compared with its counterpart which trains an individual model for each language pair. In this work we propose the multilingual distillation framework to boost the accuracy of multilingual NMT, so as to match or even surpass the accuracy of individual models.

### 2.3 KNOWLEDGE DISTILLATION

The early adoption of knowledge distillation is for model compression (Bucilu et al., 2006), where the goal is to deliver a compact student model that matches the accuracy of a large teacher model or the ensemble of multiple models. Knowledge distillation has soon been applied to a variety of tasks, including image classification (Hinton et al., 2015; Furlanello et al., 2018; Yang et al., 2018; Anil et al., 2018; Li et al., 2017), speech recognition (Hinton et al., 2015) and natural language processing (Kim & Rush, 2016a; Freitag et al., 2017). Recent works (Furlanello et al., 2018; Yang et al., 2018) even demonstrate that student model can surpass the accuracy of the teacher model, even if the teacher model is of the same capacity as the student model. Zhang et al. (2017) propose the mutual learning to enable multiple student models to learn collaboratively and teach each other by knowledge distillation, which can improve the accuracy of those individual models. Anil et al. (2018) propose online distillation to improve the scalability of distributed model training and the training accuracy.

In this paper, we develop the multilingual distillation framework for multilingual NMT. Our work differs from Zhang et al. (2017) and Anil et al. (2018) in that they collaboratively train multiple student models with codistillation, while we use multiple teacher models to train a single student model, the multilingual NMT model.

## 3 METHOD

As mentioned, when there are many language pairs and each pair has enough training data, the accuracy of individual models for those language pairs is usually higher than that of the multilingual model, given that the multilingual model has limited capacity comparing with the sum of all the individual models. Therefore, we propose to teach the multilingual model using the individual models as teachers. Here we first describe the idea of knowledge distillation in neural machine translation for the case of one teacher and one student, and then introduce our method in the multilingual setting with multiple teachers (the individual models) and one student (the multilingual model).

### 3.1 ONE TEACHER AND ONE STUDENT

Denote $D = \{(x, y) \in \mathcal{X} \times \mathcal{Y}\}$ as the bilingual corpus of a language pair. The log-likelihood loss (cross-entropy with one-hot label) on corpus $D$ with regard to an NMT model $\theta$ can be formulated as follows:

$$\mathcal{L}_{\text{NLL}}(D; \theta) = - \sum_{(x,y) \in D} \log P(y|x; \theta),$$

$$\log P(y|x; \theta) = \sum_{t=1}^{T_y} \sum_{k=1}^{|V|} \mathbb{1}\{y_t = k\} \log P(y_t = k|y_{<t}, x; \theta),$$

(1)

where $T_y$ is the length of the target sentence, $|V|$ is the vocabulary size of the target language, $y_t$ is the $t$-th target token, $\mathbb{1}\{\cdot\}$ is the indicator function that represents the one-hot label, and $P(\cdot|\cdot)$ is the conditional probability with model $\theta$.

In knowledge distillation, the student (with model parameter $\theta$) not only matches the outputs of the ground-truth one-hot label, but also to the probability outputs of the teacher model (with parameter $\theta_T$). Denote the output distribution of the teacher model for token $y_t$ as $Q(y_t|y_{<t}, x; \theta_T)$. The cross

entropy between two distributions serves as the distillation loss:

$$\mathcal{L}_{\text{KD}}(D; \theta, \theta_T) = -\sum_{(x,y)\in D} \sum_{t=1}^{T_y} \sum_{k=1}^{|V|} Q\{y_t = k|y_{<t}, x; \theta_T\} \log P(y_t = k|y_{<t}, x; \theta). \quad (2)$$

The difference between $\mathcal{L}_{\text{NLL}}(D; \theta)$ and $\mathcal{L}_{\text{KD}}(D; \theta, \theta_T)$ is that the target distribution of $\mathcal{L}_{\text{KD}}(D; \theta, \theta_T)$ is no longer the original one-hot label, but teacher's output distribution which is more smooth by assigning non-zero probabilities to more than one word and yields smaller variance in gradients (Hinton et al., 2015). Then the total loss function becomes

$$\mathcal{L}_{\text{ALL}}(D; \theta, \theta_T) = (1 - \lambda)\mathcal{L}_{\text{NLL}}(D; \theta) + \lambda\mathcal{L}_{\text{KD}}(D; \theta, \theta_T), \quad (3)$$

where $\lambda$ is the coefficient to trade off the two loss terms.

## 3.2 Multilingual Distillation with Multiple Teachers and One Student

Let $L$ denote the total number of language pairs in our setting, superscript $l \in [L]$ denote the index of language pair, $D^l$ denote the bilingual corpus for the $l$-th language pair, $\theta_M$ denote the parameters of the (student) multilingual model, and $\theta_I^l$ denote the parameters of the (teacher) individual model for $l$-th language pair. Therefore, $\mathcal{L}_{\text{NLL}}(D; \theta_M)$ denotes the log-likelihood loss on training data $D$, and $\mathcal{L}_{\text{ALL}}(D^l; \theta_M, \theta_I^l)$ denotes the total loss on training data $D^l$, which consists of the original log-likelihood loss and the distillation loss by matching to the outputs from the teacher model $\theta_I^l$.

The multilingual distillation process is summarized in Algorithm 1. As can be seen in Line 1, our algorithm takes pretrained individual models for each language pair as inputs. Note that those models can be pretrained using the same datasets $\{D^l\}_{l=1}^L$ or different datasets, and they can share the same network structure as the multilingual model or use different architectures. For simplification, in our experiments, we use the same datasets to pretrain the individual models and they share the same architecture as the multilingual model. In Line 8-9, the multilingual model learns from both the ground-truth data and the individual models with loss $\mathcal{L}_{\text{ALL}}$ when its accuracy has not surpassed the individual model for a certain threshold $\tau$ (which is checked in Line 15-19 every $\mathcal{T}_{\text{check}}$ steps according to the accuracy in validation set); otherwise, the multilingual model only learns from the ground-truth data using the original log-likelihood loss $\mathcal{L}_{\text{NLL}}$ (in Line 10-11).

---

**Algorithm 1** Knowledge Distillation for Multilingual NMT

---

1: **Input**: Training corpus $\{D^l\}_{l=1}^L$ and pretrained individual models $\{\theta_I^l\}_{l=1}^L$ for $L$ language pairs, learning rate $\eta$, total training steps $\mathcal{T}$, distillation check step $\mathcal{T}_{\text{check}}$, threshold $\tau$ of distillation accuracy.
2: **Initialize**: Randomly initialize multilingual model $\theta_M$. Set current training step $T = 0$, accumulated gradient $g = \mathbf{0}$, distillation flag $f^l = \text{True}$ for $l \in [L]$.
3: **while** $T < \mathcal{T}$ **do**
4:     $T = T+1$
5:     $g = \mathbf{0}$
6:     **for** $l \in [L]$ **do**
7:         Randomly sample a mini-batch of sentence pairs $(\mathbf{x}^l, \mathbf{y}^l)$ from $D^l$.
8:         **if** $f^l == \text{True}$ **do**
9:           Compute and accumulate the gradient on loss $\mathcal{L}_{\text{ALL}}((\mathbf{x}^l, \mathbf{y}^l); \theta_M, \theta_I^l)$: $g \mathrel{+}= \partial\mathcal{L}_{\text{ALL}}/\partial\theta_M$.
10:        **else**
11:          Compute and accumulate the gradient on loss $\mathcal{L}_{\text{NLL}}((\mathbf{x}^l, \mathbf{y}^l); \theta_M)$: $g \mathrel{+}= \partial\mathcal{L}_{\text{NLL}}/\partial\theta_M$.
12:        **end if**
13:     **end for**
14:     Update $\theta_M$: $\theta_M = \theta_M - \eta * g$
15:     **if** $T \% \mathcal{T}_{\text{check}} == 0$ **do**
16:        **for** $l \in [L]$ **do**
17:          **if** $\text{Accuracy}(\theta_M) < \text{Accuracy}(\theta_I^l) + \tau$ **do** $f^l = \text{True}$ **else** $f^l = \text{False}$ **end if**
18:        **end for**
19:     **end if**
20: **end while**

---

### 3.3 DISCUSSION

**Selective Distillation**  Considering that distillation from a bad teacher model is likely to hurt the student model and thus result in inferior accuracy, we selectively use distillation in the training process, as shown in Line 15-19 in Algorithm 1. When the accuracy of multilingual model surpasses the individual model for the accuracy threshold $\tau$ on a certain language pair, we remove the distillation loss and just train the model with original negative log-likelihood loss for this pair. Note that in one iteration, one language may not uses the distillation loss; it is very likely in later iterations that this language will be distilled again since the multilingual model may become worse than the teacher model for this language. Therefore, we call this mechanism as selective distillation. We also verify the effectiveness of the selective distillation in experiment part (Section 4.3).

**Top-K Distillation**  It is burdensome to load all the teacher models in the GPU memory for distillation considering there are dozens or even hundreds of language pairs in the multilingual setting. Alternatively, we first generate the output probability distribution of each teacher model for the sentence pairs offline, and then just load the top-K probabilities of the distribution into memory and normalize them so that they sum to 1 for distillation. This can reduce the memory cost again from the scale of $|V|$ (the vocabulary size) to K. We also study in Section 4.3 that top-K distribution can result in comparable or better distillation accuracy than the full distribution.

## 4 EXPERIMENTS

We test our proposed method on three public datasets: IWSLT, WMT, and Ted talk translation tasks. We first describe experimental settings, report results, and conduct some analyses on our method.

### 4.1 SETTINGS

**Datasets**  We use three datasets in our experiment. *IWSLT*: We collect 12 languages↔English translation pairs from IWSLT evaluation campaign[2] from year 2014 to 2016. *WMT*: We collect 6 languages↔English translation pairs from WMT translation task[3]. *Ted Talk*: We use the common corpus of TED talk which contains translations between multiple languages (Ye et al., 2018). We select 44 languages in this corpus that has sufficient data for our experiments. More descriptions about the three datasets can be found in Appendix (Section 1). We also list the language code according to ISO-639-1 standard[4] for the languages used in our experiments in Appendix (Section 2). All the sentences are first tokenized with moses tokenizer[5] and then segmented into subword symbols using Byte Pair Encoding (BPE) (Sennrich et al., 2016). We learn the BPE merge operations across all the languages and keep the output vocabulary of the teacher and student model the same, to ensure knowledge distillation.

**Model Configurations**  We use the Transformer (Vaswani et al., 2017) as the basic NMT model structure since it achieves state-of-the-art accuracy and becomes a popular choice for recent NMT researches. We use the same model configuration for individual models and the multilingual model. For IWSLT and Ted talk tasks, the model hidden size $d_{\text{model}}$, feed-forward hidden size $d_{\text{ff}}$, number of layer are 256, 1024 and 2, while for WMT task, the three parameters are 512, 2048 and 6 respectively considering its large scale of training data.

**Training and Inference**  For the multilingual model training, we up sample the data of each language to make all languages have the same size of data. The mini batch size is set to roughly 8192 tokens. We train the individual models with 4 NVIDIA Tesla V100 GPU cards and multilingual models with 8 of them. We follow the default parameters of Adam optimizer (Kingma & Ba, 2014) and learning rate schedule in Vaswani et al. (2017). For the individual models, we use 0.2 dropout, while for multilingual models, we use 0.1 dropout according to the validation performance. For

---

[2]https://wit3.fbk.eu/

[3]http://www.statmt.org/wmt16/translation-task.html, http://www.statmt.org/wmt17/translation-task.html

[4]https://www.loc.gov/standards/iso639-2/php/code_list.php

[5]https://github.com/moses-smt/mosesdecoder/blob/master/scripts/tokenizer/tokenizer.perl

knowledge distillation, we set $\mathcal{T}_{check} = 3000$ steps (nearly two training epochs), the accuracy threshold $\tau = 1$ BLEU score, the distillation coefficient $\lambda = 0.5$ and the number of teacher's outputs $K = 8$ according to the validation performance. During inference, we decode with beam search and set beam size to 4 and length penalty $\alpha = 1.0$ for all the languages. We evaluate the translation quality by tokenized case sensitive BLEU (Papineni et al., 2002) with multi-bleu.pl[6]. Our codes are implemented based on fairseq[7] and we will release the codes once the paper is published.

| Language | Individual | Multi-Baseline | Multi-Distillation | Δ |
|---|---|---|---|---|
| Ar→En | 31.19 | 29.24 (-1.95) | 31.25 (+0.06) | +2.01 |
| Cs→En | 28.04 | 26.09 (-1.95) | 27.09 (-0.95) | +1.00 |
| De→En | 33.07 | 32.74 (-0.33) | 34.02 (+0.95) | +1.28 |
| He→En | 37.42 | 35.18 (-2.24) | 37.33 (-0.09) | +2.15 |
| Nl→En | 35.94 | 36.54 (+0.60) | 37.69 (+1.75) | +1.15 |
| Pt→En | 44.30 | 43.49 (-0.81) | 44.69 (+0.39) | +1.20 |
| Ro→En | 36.92 | 36.41 (-0.51) | 38.01 (+1.09) | +1.60 |
| Ru→En | 23.04 | 23.12 (+0.08) | 23.76 (+0.72) | +0.64 |
| Th→En | 18.24 | 19.33 (+1.09) | 19.90 (+1.66) | +0.57 |
| Tr→En | 22.74 | 22.42 (-0.32) | 23.75 (+1.01) | +1.33 |
| Vi→En | 26.06 | 26.37 (+0.31) | 27.04 (+0.98) | +0.67 |
| Zh→En | 19.44 | 18.82 (-0.62) | 19.52 (+0.08) | +0.70 |

Table 1: BLEU scores of 12 languages→English on the IWLST dataset. The BLEU scores in () represent the difference between the multilingual model and individual models. Δ represents the improvements of our multi-distillation method over the multi-baseline.

| Language | Individual | Multi-Baseline | Multi-Distillation | Δ |
|---|---|---|---|---|
| En→Ar | 13.67 | 12.73 (-0.94) | 13.80 (+0.13) | +1.07 |
| En→Cs | 17.81 | 17.33 (-0.48) | 18.69 (+0.88) | +1.37 |
| En→De | 26.13 | 25.16 (-0.97) | 26.76 (+0.63) | +1.60 |
| En→He | 24.15 | 22.73 (-1.42) | 24.42 (+0.27) | +1.69 |
| En→Nl | 30.88 | 29.51 (-1.37) | 30.52 (-0.36) | +1.01 |
| En→Pt | 37.63 | 35.93 (-1.70) | 37.23 (-0.40) | +1.30 |
| En→Ro | 27.23 | 25.68 (-1.55) | 27.11 (-0.12) | +1.42 |
| En→Ru | 17.40 | 16.26 (-1.14) | 17.42 (+0.02) | +1.16 |
| En→Th | 26.45 | 27.18 (+0.73) | 27.62 (+1.17) | +0.45 |
| En→Tr | 12.47 | 11.63 (-0.84) | 12.84 (+0.37) | +1.21 |
| En→Vi | 27.88 | 28.04 (+0.16) | 28.69 (+0.81) | +0.65 |
| En→Zh | 10.95 | 10.12 (-0.83) | 10.41 (-0.54) | +0.29 |

Table 2: BLEU scores of English→12 languages on the IWLST dataset. The BLEU scores in () represent the difference between the multilingual model and individual models. Δ represents the improvements of our multi-distillation method over the multi-baseline.

## 4.2 RESULTS

**Results on IWSLT** Multilingual NMT usually consists of three settings: many-to-one, one-to-many and many-to-many. As many-many translation can be bridged though many-to-one and one-to-many setting, we just conduct the experiments on many-to-one and one-to-many settings. We first show the results of 12 languages→English translations on the IWLST dataset are shown in Table 1. There are 3 methods for comparison: 1) *Individual*, each language pair with a separate model; 2) *Multi-Baseline*, the baseline multilingual model, simply training all the language pairs in one model; 3) *Multi-Distillation*, our multilingual model with knowledge distillation. We have several observations. First, the multilingual baseline performs worse than individual models on most languages. The only exception is the languages with small training data, which benefit from data augmentation in multilingual training. Second, our method outperforms the multilingual baseline for all the languages, demonstrating the effectiveness of our framework for multilingual NMT. More importantly,

---

[6]https://github.com/moses-smt/mosesdecoder/blob/master/scripts/generic/multi-bleu.perl
[7]https://github.com/pytorch/fairseq

compared with the individual models, our method achieves similar or even better accuracy (better on 10 out of 12 languages), with only 1/12 model parameters of the sum of all individual models.

One-to-many setting is usually considered as more difficult than many-to-one setting, as it contains different target languages which is hard to handle. Here we show how our method performs in one-to-many setting in Table 2. It can be seen that our method can maintain the accuracy (even better on most languages) compared with the individual models. We still improve over the multilingual baseline by nearly 1 BLEU score, which demonstrates the effectiveness of our method.

| Language | *Individual* | *Multi-Baseline* | *Multi-Distillation* | $\Delta$ |
|----------|------------|----------------|--------------------|----------|
| Cs-En | 25.29 | 23.82 (-1.47) | 25.37 (+0.08) | +1.55 |
| De-En | 34.44 | 34.21 (-0.23) | 36.22 (+1.78) | +2.01 |
| Fi-En | 21.23 | 22.99 (+1.76) | 24.32 (+3.09) | +1.33 |
| Lv-En | 16.26 | 16.25 (-0.01) | 18.43 (+2.17) | +2.18 |
| Ro-En | 35.81 | 35.04 (-0.77) | 36.51 (+0.70) | +1.47 |
| Ru-En | 29.39 | 28.92 (-0.47) | 30.82 (+1.43) | +1.90 |

Table 3: BLEU scores of 6 languages→English on the WMT dataset. The BLEU scores in () represent the difference between the multilingual model and individual models. $\Delta$ represents the improvements of our multi-distillation method over the multi-baseline.

| Language | *Individual* | *Multi-Baseline* | *Multi-Distillation* | $\Delta$ |
|----------|------------|----------------|--------------------|----------|
| En-Cs | 22.58 | 21.39 (-1.19) | 23.10 (+0.62) | +1.81 |
| En-De | 31.40 | 30.08 (-1.32) | 31.42 (+0.02) | +1.34 |
| En-Fi | 22.08 | 19.52 (-2.56) | 21.56 (-0.52) | +2.04 |
| En-Lv | 14.92 | 14.51 (-0.41) | 15.32 (+0.40) | +0.81 |
| En-Ro | 31.67 | 29.88 (-1.79) | 31.39 (-0.28) | +1.51 |
| En-Ru | 24.36 | 22.96 (-1.40) | 24.02 (-0.34) | +1.06 |

Table 4: BLEU scores of English→ 6 languages on the WMT dataset.

**Results on WMT** The results of 6 languages→English translations on the WMT dataset are reported in Table 3. It can be seen that the multi-baseline model performs worse than the individual models on 5 out of 6 languages, while in contrast, our method performs better on all the 6 languages. Particularly, our method improves the accuracy of some languages with more than 2 BLEU scores over individual models. The results of one-to-many setting on WMT dataset are reported in Table 4. It can be seen that our method outperforms the multilingual baseline by more than 1 BLEU score on nearly all the languages.

| Language | Ar | Bg | Cs | Da | De | El | Es | Et | Fa | Fi | Frca |
|----------|------|------|------|------|------|------|------|------|------|------|------|
| $\Delta_1$ | -1.50 | -9.46 | 1.88 | 4.02 | -0.10 | 0.80 | 0.23 | 8.20 | 0.09 | 6.44 | 15.8 |
| $\Delta_2$ | 1.73 | 1.42 | 1.13 | 1.82 | 1.68 | 1.45 | 1.63 | 0.77 | 1.83 | 1.10 | 1.24 |
| Language | Fr | Gl | He | Hi | Hr | Hu | Hy | Id | It | Ja | Ka |
| $\Delta_1$ | 0.13 | 19.26 | -1.59 | 10.16 | 1.46 | -0.11 | 8.87 | 1.36 | -0.56 | -0.03 | 11.20 |
| $\Delta_2$ | 1.48 | 1.58 | 2.26 | 1.07 | 1.21 | 1.80 | 0.92 | 1.48 | 1.48 | 0.95 | 1.55 |
| Language | Ko | Ku | Lt | Mk | My | Nb | Nl | Pl | Ptbr | Pt | Ro |
| $\Delta_1$ | -0.42 | 7.75 | 4.46 | 10.72 | 7.63 | 14.07 | -0.20 | 1.32 | 0.13 | 8.76 | 0.66 |
| $\Delta_2$ | 1.43 | 1.55 | 1.69 | 0.80 | 1.31 | 1.47 | 1.68 | 0.80 | 1.45 | 1.98 | 1.70 |
| Language | Ru | Sk | Sl | Sq | Sr | Sv | Th | Tr | Uk | Vi | Zh |
| $\Delta_1$ | 0.65 | 4.23 | 11.87 | 5.03 | 1.58 | 2.39 | 1.17 | -0.79 | 2.04 | 0.15 | 6.83 |
| $\Delta_2$ | 0.99 | 0.93 | 1.15 | 1.68 | 1.44 | 1.00 | 0.62 | 1.88 | 0.98 | 0.77 | 0.58 |

Table 5: BLEU scores improvements of our method over the individual models ($\Delta_1$) and multi-baseline model ($\Delta_2$) on the 44 languages→English in the Ted talk dataset.

**Results on Ted Talk** Now we study the effectiveness of our method on a large number of languages. The experiments are conducted on the 44 languages→English on the Ted talk dataset. Due to the large number of languages and space limitations, we just show the BLEU score improvements of our method over individual models and the multi-baseline for each language in Table 5, and leave the detailed experiment results to Appendix (Section 3). It can be seen that our method can improve over the multi-baseline for all the languages, mostly with more than 1 BLEU score improvements. Our method can also match or even surpass individual models for most languages, not to mention that the number of parameters of our method is only $1/44$ of that of the sum of 44 individual models. Our method achieves larger improvements on some languages, such as Da, Et, Fi, Hi and Hy, than others. We find this is correlated with the data size of the languages, which are listed in Appendix (Table 13). When a language is of smaller data size, it may get more improvement due to the benefit of multilingual training.

### 4.3 ANALYSIS

In this section, we conduct thorough analyses on our proposed method for multilingual NMT.

**Selective Distillation** We study the effectiveness of the selective distillation (discussed in Section 3.3) on the Ted talk dataset, as shown in Table 6. We list the 16 languages on which the two methods (selective distillation, and distillation all the time) that have difference bigger than 0.5 in terms of BLEU score. It can be seen that selective distillation performs better on 13 out of 16 languages, with large BLEU score improvements, which demonstrates the effectiveness of the selective distillation.

|                           | Bg    | Et    | Fi    | Fr    | Gl    | Hi    | Hy    | Ka    |
|---------------------------|-------|-------|-------|-------|-------|-------|-------|-------|
| distillation all the time | 28.07 | 12.64 | 15.13 | 33.69 | 30.28 | 18.86 | 19.88 | 14.04 |
| selective distillation    | 29.18 | 15.63 | 17.23 | 34.32 | 31.90 | 21.00 | 21.17 | 18.27 |
| $\Delta$                  | +1.11 | +2.99 | +2.10 | +0.63 | +1.62 | +2.14 | +1.29 | +4.23 |
|                           | Ku    | Mk    | My    | Sl    | Zh    | Pl    | Sk    | Sv    |
| distillation all the time | 8.50  | 32.10 | 14.02 | 22.10 | 17.22 | 25.05 | 30.45 | 37.88 |
| selective distillation    | 13.38 | 32.65 | 15.17 | 23.68 | 19.39 | 24.30 | 29.91 | 36.92 |
| $\Delta$                  | +4.88 | +0.55 | +1.15 | +1.58 | +2.17 | -0.75 | -0.54 | -0.96 |

Table 6: BLEU scores of selective distillation (our method) and distillation all the time during the training process on the Ted talk dataset.

**Top-K Distillation** In our experiments, the student model just matches the top-K output distribution of the teacher model, instead of the full distribution, in order to reduce the memory cost. We analyze whether there is accuracy difference between the top-K distribution and the full distribution. We conduct experiments on IWSLT dataset with varying $K$ (from 1 to $|V|$, where $|V|$ is the vocabulary size), and just show the BLEU scores on the validation set of De-En translation due to space limitation, as illustrated in Table 7. It can be seen that increasing $K$ from 1 to 8 will improve the accuracy, while bigger $K$ will bring no gains, even with the full distribution ($K = |V|$).

| Top-K | 1     | 2     | 4     | 8     | 16    | 32    | 64    | 128   | $|V|$ |
|-------|-------|-------|-------|-------|-------|-------|-------|-------|-------|
| BLEU  | 33.45 | 33.86 | 34.47 | 34.76 | 34.66 | 34.68 | 34.54 | 34.47 | 34.49 |

Table 7: BLEU scores on De-En translation with varying Top-K distillation on the IWSLT dataset.

**Back Distillation** In our current distillation algorithm, we fix the individual models and use them to teach and improve the multilingual model. After such a distillation process, the multilingual model outperforms the individual models on most of the languages. Then naturally, we may wonder whether this improved multilingual model can further be used to teach and improve individual models through knowledge distillation. We call such a process back distillation. We conduct the experiments on the IWSLT dataset, and find that the accuracy of 9 out of 12 languages gets improved,

| Language | Ar | Cs | De | Nl | Ro | Ru | Th | Tr | Vi |
|---|---|---|---|---|---|---|---|---|---|
| Individual | 31.19 | 28.04 | 33.07 | 35.94 | 36.92 | 23.04 | 18.24 | 22.74 | 26.06 |
| +Back Distillation | 31.39 | 29.44 | 33.71 | 36.86 | 37.28 | 23.36 | 19.42 | 23.58 | 27.17 |
| Δ | +0.20 | +1.40 | +0.64 | +0.92 | +0.36 | +0.32 | +1.18 | +0.84 | +1.11 |

Table 8: BLEU score improvements of the individual models with back distillation on the IWSLT dataset.

as shown in Table 10. The other 3 languages (He, Pt, Zh) cannot get improvements because the improved multilingual model performs very close to individual models, as shown in Table 1.

**Comparison with Sequence-Level Knowledge Distillation**    We conduct experiments to compare the word-level knowledge distillation (the exact method used in our paper) with sequence-level knowledge distillation(Kim & Rush, 2016b) on IWSLT dataset. As shown in Table 9, sequence-level knowledge distillation results in consistently inferior accuracy on all languages compared with word-level knowledge distillation used in our work.

| Language | Sequence-level | Word-level (Our Method) | Δ |
|---|---|---|---|
| En-Ar | 12.79 | 13.80 | 1.01 |
| En-Cs | 17.01 | 18.69 | 1.68 |
| En-De | 25.89 | 26.76 | 0.87 |
| En-He | 22.92 | 24.42 | 1.50 |
| En-Nl | 29.99 | 30.52 | 0.53 |
| En-Pt | 36.12 | 37.23 | 1.10 |
| En-Ro | 25.75 | 27.11 | 1.36 |
| En-Ru | 16.38 | 17.42 | 1.04 |
| En-Th | 27.52 | 27.62 | 0.10 |
| En-Tr | 11.11 | 12.84 | 1.73 |
| En-Vi | 28.08 | 28.69 | 0.61 |
| En-Zh | 10.25 | 10.41 | 0.16 |

Table 9: BLEU scores of sequence-level knowledge distillation and word-level knowledge distillation on the IWSLT dataset.

| Language | Ar | Cs | De | Nl | Ro | Ru | Th | Tr | Vi |
|---|---|---|---|---|---|---|---|---|---|
| Individual | 31.19 | 28.04 | 33.07 | 35.94 | 36.92 | 23.04 | 18.24 | 22.74 | 26.06 |
| +Back Distillation | 31.39 | 29.44 | 33.71 | 36.86 | 37.28 | 23.36 | 19.42 | 23.58 | 27.17 |
| Δ | +0.20 | +1.40 | +0.64 | +0.92 | +0.36 | +0.32 | +1.18 | +0.84 | +1.11 |

Table 10: BLEU score improvements of the individual models with back distillation on the IWSLT dataset.

**Generalization Analysis**    Previous works (Yang et al., 2018; Lan et al., 2018) have shown that knowledge distillation can help a model generalize well to unseen data, and thus yield better performance. We analyze how distillation in multilingual setting helps the model generalization. Previous studies (Keskar et al., 2016; Chaudhari et al., 2016) demonstrate the relationship between model generalization and the width of local minima in loss surface. Wider local minima can make the model more robust to small perturbations in testing. Therefore, we compare the generalization capability of the two multilingual models (our method and the baseline) by perturbing their parameters.

Specifically, we perturb a model $\theta$ as $\theta_i(\sigma) = \theta_i + \bar{\theta} * \mathcal{N}(0, \sigma^2)$, where $\theta_i$ is the $i$-th parameter of the model, $\bar{\theta}$ is the average of all the parameters in $\theta$. We sample from the normal distribution $\mathcal{N}$ with standard variance $\sigma$ and larger $\sigma$ represents bigger perturbation on the parameter. We conduct the analyses on the IWSLT dataset and vary $\sigma \in [0.05, 0.1, 0.15, 0.2, 0.25, 0.3]$. Figure 1a shows the loss curve in the test set with varying $\sigma$. As can be seen, while both the two losses increase with the

increase of $\sigma$, the loss of the baseline model increases quicker than our method. We also show three test BLEU curves on three translation pairs (Figure 1b: Ar-En, Figure 1c: Cs-En, Figure 1d: De-En, which are randomly picked from the 12 languages pairs on the IWSLT dataset). We observe that the BLEU score of the multilingual baseline drops quicker than our method, which demonstrates that our method helps the model find wider local minima and thus generalize better.

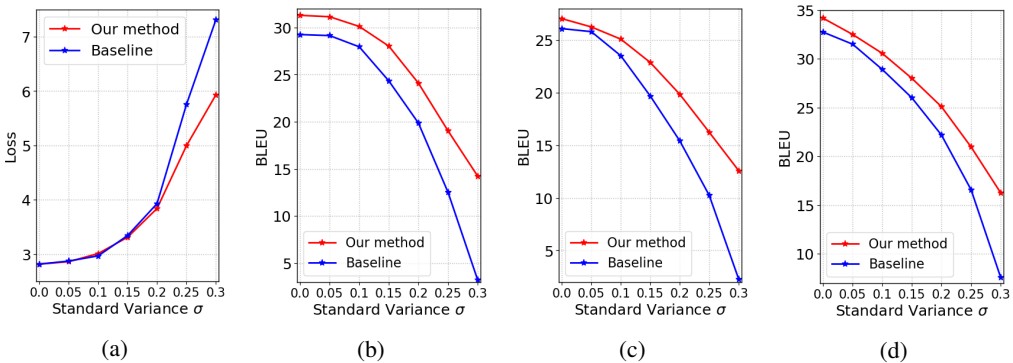

Figure 1: The loss (Figure a) and BLEU score (Figure b: Ar-En, Figure c: Cs-En, Figure d: De-En) changes on the test set of the IWSLT dataset, with varying perturbation parameter $\sigma$.

## 5 CONCLUSION

In this work, we have proposed a distillation-based approach to boost the accuracy of multilingual NMT, which is usually of lower accuracy than the individual models in previous works. Experiments on three translation datasets with up to 44 languages demonstrate the multilingual model based on our proposed method can nearly match or even outperform the individual models, with just $1/N$ model parameters (N is up to 44 in our experiments).

In the future, we will conduct more deep analyses about how distillation helps the multilingual model training. We will apply our method to larger datasets and more languages pairs (hundreds or even thousands), to study the upper limit of our proposed method.

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

APPENDIX

## 1 DATASET DESCRIPTION

We give a detailed description about the *IWSLT*,*WMT* and *Ted Talk* datasets used in experiments.

*IWSLT*: We collect 12 languages↔English translation pairs from IWSLT evaluation campaign[8] from year 2014 to 2016. Each language pair contains roughly 80K to 200K sentence pairs. We use the official validation and test sets for each language pair. The data sizes of the training set for each language↔English pair are listed in Table 11.

| Language | Ar | Cs | De | He | Nl | Pt |
|---|---|---|---|---|---|---|
| Training Data | 174K | 114K | 167K | 180K | 174K | 167K |
| Language | Ro | Ru | Th | Tr | Vi | Zh |
| Training Data | 177K | 173K | 83K | 150K | 131K | 209K |

Table 11: The training data size on the 12 languages↔ English on the IWSLT dataset.

*WMT*: We collect 6 languages↔English translation pairs from WMT translation task[9]. We use 5 language↔English translation pairs from WMT 2016 dataset: Cs-En, De-En, Fi-En, Ro-En, Ru-En and one other translation pair from WMT 2017 dataset: Lv-En. We use the official released validation and test sets for each language pair. The training data sizes of each language↔English pair are shown in the Table 12.

| Language | Cs | De | Fi | Lv | Ro | Ru |
|---|---|---|---|---|---|---|
| Training Data | 1.0M | 4.5M | 2.5M | 4.5M | 2.2M | 2.1M |

Table 12: The training data size on the 6 languages↔English on the WMT dataset.

*Ted Talk*: We use the common corpus of TED talk which contains translations between multiple languages (Ye et al., 2018)[10]. We select 44 languages in this corpus that has sufficient data for our experiments. We use the official validation and test sets for each language pair. The data sizes of the training set for each language↔English pair are listed in Table 13.

| Language | Ar | Bg | Cs | Da | De | El | Es | Et | Fa | Fi | Frca |
|---|---|---|---|---|---|---|---|---|---|---|---|
| Training Data | 214K | 174k | 103k | 45k | 168k | 134k | 196k | 11k | 151k | 24k | 20k |
| Language | Fr | Gl | He | Hi | Hr | Hu | Hy | Id | It | Ja | Ka |
| Training Data | 192K | 10K | 212K | 19K | 122K | 147K | 21K | 87K | 205K | 204K | 13K |
| Language | Ko | Ku | Lt | Mk | My | Nb | Nl | Pl | Ptbr | Pt | Ro |
| Training Data | 206K | 10K | 42K | 25K | 21K | 16K | 184K | 176K | 185K | 52K | 180K |
| Language | Ru | Sk | Sl | Sq | Sr | Sv | Th | Tr | Uk | Vi | Zh |
| Training Data | 208K | 61K | 20K | 45K | 137K | 57K | 98K | 182K | 108K | 172K | 200K |

Table 13: The training data size on the 44 languages↔ English on the Ted talk dataset.

## 2 LANGUAGE NAME AND CODE

The language names and their corresponding language codes according to ISO 639-1 standard[11] are listed in Table 14.

---

[8]https://wit3.fbk.eu/

[9]http://www.statmt.org/wmt16/translation-task.html, http://www.statmt.org/wmt17/translation-task.html

[10]https://github.com/neulab/word-embeddings-for-nmt

[11]https://www.loc.gov/standards/iso639-2/php/code_list.php

| Language | Code | Language | Code | Language | Code | Language | Code |
|---|---|---|---|---|---|---|---|
| Arabic | Ar | Bulgarian | Bg | Czech | Cs | Danish | Da |
| German | De | Greek | El | English | En | Spanish | Es |
| Persian | Fa | Finnish | Fi | French | Fr | Galician | Gl |
| Hebrew | He | Hindi | Hi | Croatian | Hr | Hungarian | Hu |
| Armenian | Hy | Indonesian | Id | Italian | It | Japanese | Ja |
| Georgian | Ka | Korean | Ko | Kurdish | Ku | Lithuanian | Lt |
| Latvian | Lv | Macedonian | Mk | Burmese | My | Norwegian | Nb |
| Dutch | Nl | Polish | Pl | Portuguese | Pt | Romanian | Ro |
| Russian | Ru | Slovak | Sk | Slovenian | Sl | Albanian | Sq |
| Serbian | Sr | Swedish | Sv | Thai | Th | Turkish | Tr |
| Ukrainian | Uk | Vietnamese | Vi | Chinese | Zh | | |

Table 14: The ISO 639-1 code of each language in our experiments. There are two extra language codes in our datasets: Ptbr represents Portuguese spoken in Brazil, Frca represents French spoken in Canada.

## 3 Results on Ted Talk Dataset

The detailed results of the 44 languages→English on the Ted talk dataset are listed in Table 15. It can be seen that while multilingual baseline performs worse than the individual model, multilingual model based on our method nearly matches and even outperforms the individual model. Note that the multilingual model handles 44 languages in total, which means our method can reduce the model parameters size to $1/44$ without loss of accuracy.

| Language | Ar | Bg | Cs | Da | De | El | Es | Et | Fa |
|---|---|---|---|---|---|---|---|---|---|
| *Individual* | 31.07 | 38.64 | 26.42 | 38.21 | 34.63 | 36.69 | 41.20 | 7.43 | 26.67 |
| *Multilingual (Baseline)* | 27.84 | 27.76 | 27.17 | 40.41 | 32.85 | 36.04 | 39.80 | 14.86 | 24.93 |
| *Multilingual (Our method)* | 29.57 | 29.18 | 28.30 | 42.23 | 34.53 | 37.49 | 41.43 | 15.63 | 26.76 |

| Language | Fi | Frca | Fr | Gl | He | Hi | Hr | Hu | Hy |
|---|---|---|---|---|---|---|---|---|---|
| *Individual* | 10.78 | 18.52 | 39.62 | 12.64 | 36.81 | 10.84 | 34.14 | 24.67 | 12.30 |
| *Multilingual (Baseline)* | 16.12 | 33.08 | 38.27 | 30.32 | 32.96 | 19.93 | 34.39 | 22.76 | 20.25 |
| *Multilingual (Our method)* | 17.22 | 34.32 | 39.75 | 31.9 | 35.22 | 21.00 | 35.6 | 24.56 | 21.17 |

| Language | Id | It | Ja | Ka | Ko | Ku | Lt | Mk | My |
|---|---|---|---|---|---|---|---|---|---|
| *Individual* | 29.20 | 38.06 | 13.31 | 7.06 | 18.54 | 5.63 | 18.19 | 21.93 | 7.53 |
| *Multilingual (Baseline)* | 29.08 | 36.02 | 12.33 | 16.71 | 16.71 | 11.83 | 20.96 | 31.85 | 13.85 |
| *Multilingual (Our method)* | 30.56 | 37.50 | 13.28 | 18.26 | 18.14 | 13.38 | 22.65 | 32.65 | 15.16 |

| Language | Nb | Nl | Pl | Ptbr | Pt | Ro | Ru | Sk | Sl |
|---|---|---|---|---|---|---|---|---|---|
| *Individual* | 27.28 | 35.85 | 22.98 | 44.28 | 33.81 | 34.07 | 24.36 | 25.67 | 11.80 |
| *Multilingual (Baseline)* | 39.88 | 33.97 | 23.50 | 42.96 | 40.59 | 33.03 | 24.02 | 28.97 | 22.52 |
| *Multilingual (Our method)* | 41.35 | 35.65 | 24.30 | 44.41 | 42.57 | 34.73 | 25.01 | 29.90 | 23.67 |

| Language | Sq | Sr | Sv | Th | Tr | Uk | Vi | Zh | |
|---|---|---|---|---|---|---|---|---|---|
| *Individual* | 29.70 | 32.13 | 34.53 | 20.95 | 24.46 | 25.76 | 26.38 | 12.56 | |
| *Multilingual (Baseline)* | 33.05 | 32.27 | 35.92 | 21.50 | 21.79 | 26.82 | 25.76 | 18.81 | |
| *Multilingual (Our method)* | 34.73 | 33.71 | 36.92 | 22.12 | 23.67 | 27.80 | 26.53 | 19.39 | |

Table 15: BLEU scores of the individual and multilingual models on the 44 languages→English on the Ted talk dataset.

