# OpenReview forum: "Multilingual Neural Machine Translation with Knowledge Distillation"
_ICLR.cc/2019/Conference_

### Official Review · AnonReviewer2 · 2018-11-02
**Solid experimentation but...**

**Rating:** 7
**Confidence:** 4

**Review:**

... I would have liked to see some more insights.

The authors present a method for distilling knowledge from individual models to train a multilingual model. The motivation stems from the fact that while most s-o-t-a multilingual models are compact (as compared to k individual models) they fall short of the performance of the individual models. The authors demonstrate that using knowledge distillation, the performance of the multilingual model can actually be better than the individual models.

Please find below my comments and questions.

1) The authors have done a commendable job of validating their hypothesis on multiple datasets. Solid experimentation is definitely the main strength of this paper.

2) However, this strength also makes way for a weakness. The entire experimental section is just filled with tables and numbers. The same message is repeated across these multiple tables (multi+distill > single > multi). Beyond this message there are no other insights. For example,

- How does the performance depend on the divergence between source and target language?
- Why is there more important on some languages and less on others ?
- Why are the improvements on the TED dataset so much higher as compared to the other 2 datasets.
- What happens when the target language is something other than English? All the experiments report results from X-->English, why not in the other direction? The model then is not really "completely" multilingual. It is multi-source-->single target.
- Can you comment on the total training time ?
- What happens when you do not stop the distillation even when the accuracy of the student crosses that of the teachers ? What do you mean by accuracy here? Only later when you mention that \threshold = 1 BLEU it became clear that accuracy means BLEU in this context ?

3) Is it all worth it? One disappointing factor is that end of all this effort where you train K individual models and one monolithic model with distillation, the performance gain for most language pairs is really marginal (except on the TED dataset). I wonder if the same improvements could have been obtained by even more carefully fine tuning the baseline models itself.

4) On a positive note, I like the back-distillation idea and the experiments on top-K distillation

+++++++++++++++++++
I have updated my rating after reading author's responses

---

> ### Author Response · Authors · 2018-11-22
> **rebuttal from authors [1/2]**
>
> We thank Reviewer 2 for the reviews and comments! Here are our responses to the comments.
>
> Question1: How does the divergence between the source and target language influence the performance, and why is there more important on some languages and less on others?
> Answer1: We found that the performance gains of our method correlate with the training data size of each language. If a language has small training data, then it is likely to get more improvement due to the benefit of multilingual training.
>
> Question2: Why are the improvements on the TED dataset so much higher?
> Answer2: We guess you meant the improvements of our method over individual models on the TED dataset. Some languages in the TED dataset are of small data size, and thus they get higher improvement from multilingual training, by leveraging the training data from other (similar) languages.
>
> Question3: What happens when the target language is something other than English?
> Answer3: We have provided the English-to-many translation results on the IWSLT dataset in the table below. The BLEU scores in () represent the difference between the multilingual model and individual models. Delta represents the improvements of our multi-distillation method over the multi-baseline. We can see our method consistently outperforms the multilingual baseline on all languages, and can nearly match or even surpass the accuracy of the individual models, even if one-to-many translation is considered harder than many-to-one translation. We have also updated the results in the paper.  We will provide more results in the following days.
> ------------------------------------------------------------------------------------------------------------
> Language  |  Individual  |  Multilingual-baseline  |  Multilingual-distill  | Delta
> en-ar          |      13.67       |       12.73 (-0.94)             |       13.80 (0.13) 	      |  1.07
> en-cs          |      17.81       |       17.33 (-0.48)	        |       18.69 (0.88) 	      |  1.37
> en-de         |      26.13       |        25.16 (-0.97)	        |       26.76 (0.63) 	      |  1.60
> en-he         |      24.15       |        22.73 (-1.42)	        |       24.42 (0.27)	      |  1.69
> en-nl          |      30.88       |        29.51 (-1.37)            |       30.52 (-0.36)	      |  1.01
> en-pt          |      37.63       |        35.93 (-1.70)	        |       37.23 (-0.40)	      |  1.30
> en-ro          |      27.23       |        25.68 (-1.55)	        |       27.11 (-0.12)	      |  1.42
> en-ru          |     17.40        |       16.26 (-1.14)	        |       17.42 (0.02) 	      |  1.16
> en-th          |     26.45        |        27.18 (0.73) 	        |       27.62 (1.17) 	      |  0.45
> en-tr           |     12.47        |       11.63 (-0.84)	        |       12.84 (0.37) 	      |  1.21
> en-vi           |     27.88        |       28.04 (0.16) 	        |       28.69 (0.81) 	      |  0.65
> en-zh          |    10.95         |       10.12 (-0.83)	        |       10.41 (-0.54)       |  0.29
> -------------------------------------------------------------------------------------------------------------

---

> > ### Author Response · Authors · 2018-11-22
> > **rebuttal from authors [2/2]**
> >
> >
> > Question4: Can you comment on the total training time?
> > Answer4: The individual models need to be pre-trained, which will incur additional time. According to the training time statistics on IWSLT dataset with NVIDIA V100 GPU, it takes nearly 4 hours to train the individual model with 1 GPU. The total GPU time is 4hours *12 GPUs for 12 languages. The training time for multilingual baseline is nearly 11hours * 4GPUs, while our method is nearly 13 hours*4GPUs. Our method takes extra 2hours*4GPUs for the multilingual training and 4 hours*12GPUs for the individual model pretraining. Furthermore, we can assume the individual models are pre-given, which is reasonable because the production system usually wants to adapt the individual translation into multilingual setting, at the benefit of saving maintenance cost while with no accuracy degradation or even with accuracy improvement, which is exactly the goal of this work.
> >
> > Question5: What happens when you do not stop the distillation?
> > Answer5: We have shown the results when we do not stop the distillation in the submitted version (now Table 5 in the updated version). The performance of most of the languages will get worse.
> >
> > Question6: The performance gain.
> > Answer6: We’d like to point out that our goal is to train multiple languages in one model, without performance degradation compared with individual models. Our method actually outperforms individual models on many languages, which is a good byproduct. Furthermore, on the IWSLT/WMT/Ted datasets, we outperform the multilingual baseline on most languages by more than 1 BLEU score, and some languages by more than 2 BLEU scores. These are very good improvements for neural machine translation as stated in previous works [1][2].
> >
> > [1] Gehring, Jonas, et al. "Convolutional sequence to sequence learning." ICML 2017.
> > [2] Vaswani, Ashish, et al. "Attention is all you need." NIPS 2017.

---

> > > ### Comment · AnonReviewer2 · 2018-11-23
> > > **Thank you for your responses**
> > >
> > > I found most of them to be satisfying. One additional Q though: Does the language family have no impact on the performance? Your answer to Q1 suggests that everything depends only on the training data.

---

> > > > ### Author Response · Authors · 2018-11-25
> > > > **response from authors**
> > > >
> > > > Thanks for your comments!
> > > >
> > > > We found language family has smaller influence than the training data size on the gains of our multilingual distillation method. However, language family can have influence on the performance of multilingual model training. According to our previous studies, if the languages from the same language family are trained together using the multilingual baseline method, the accuracy will still drop compared with the individual model training, but the accuracy drop is less than that training the languages from different families, since one language may benefit from the data from similar languages.

---

> > > > > ### Author Response · Authors · 2018-12-13
> > > > > **Thanks for your attention**
> > > > >
> > > > > Thanks for the detailed comments. We believe we have addressed your concerns and clarified your points in the rebuttal. Do you have an updated assessment of our paper? Thanks for your consideration.

---

### Official Review · AnonReviewer1 · 2018-11-02
**Straightforward, effective technique for improving multilingual NMT, some experiments missing.**

**Rating:** 7
**Confidence:** 4

**Review:**

Summary: Train a multilingual NMT system using the technique of Johnson et al (2017), but augment the standard cross-entropy loss with a distillation component based on individual (single-language-pair) teacher models. Periodically compare the validation BLEU score of the multilingual model with that of each individual model, and turn off distillation for language pairs where the multilingual model is better. On three different corpora (IWSLT, WMT, TED) with into-English translation from numbers of source languages ranging from 6 (WMT) to 44 (TED), this technique outperforms standard distillation for every language pair, and outperforms the individual models for most language pairs. Supplementary experiments justify the strategy of selectively turning off distillation, and quantify the effect using only the top 8 vocabulary items in distillation.

The main idea makes sense, and the results are very convincing, especially since it appears that hyper-parameters were not tuned extensively (eg, weight of 0.5 on the distillation loss, for all language pairs). Implementation should be very straightforward, especially with the trick of pre-computing top-k probabilities from the teacher model at each corpus position. One small barrier to practical application that the authors fail to acknowledge is the requirement to train individual models, which will at least double training time compared to a single multilingual model.

The main missing experiment is higher-capacity multilingual models, which Johnson et al show to be beneficial in settings with a large number of language pairs. Using a multilingual model of the same (relatively small) size as the individual models as is done here is likely to be suboptimal, especially for the 44-language pair TED setting. A related point is that the corpora used seem to be quite small (eg 4.5M and 1M sentences for WMT Czech and German, respectively, while the available training corpora are closer to 15M and 4.5M). Although performance relative to individual models is still impressive - and seems to be better than than in previous work - this makes the experiments comparing to the multilingual baseline less meaningful.

Also missing are experiments on out-of-English translation, which would establish the viability of the proposed technique for many-to-many translation via bridging. Out-of-English is a more difficult problem than into-English. I can’t see any reason the proposed technique wouldn’t also work in this setting, but this remains to be shown.

Although it’s great that the technique is shown to work without embellishments, there are a few obvious strategies it would have been interesting to explore, such as making the weight on the distillation loss dependent on the difference in performance between the multilingual and individual models; and allowing for the distillation loss to be turned back on if the performance of the multilingual model starts to drift back down for a particular language pair. I also wondered about the effect of the gradient accumulation strategy in algorithm 1, where individual batches from each language pair are effectively grouped into one giant batch for the purpose of parameter updates. I can see that this could stabilize training, but it would be good to know whether it’s crucial for success, especially when the number of language pairs is large.

Further details:

As aforementioned -> As mentioned

(1) 2nd line: Doesn't make sense as written. You need to distinguish the gold
y_t from hypothesized ones in the 1() function.

Above (2): is served as -> serves as

3.2 First paragraph. Since D presumably consists of D^l for all languages l,
L_ALL(D,...) should be a function of teacher parameters theta^l for all
languages l rather than just one as written.

In top-K distillation, is the teacher distribution renormalized or simply
truncated?

Generalization analysis, pg 8: presumably you are sampling from N(0, sigma^2) -
this should be described as such.

Reference:

Johnson et al, “Google’s Multilingual Neural Machine Translation System: Enabling Zero-Shot Translation” TACL, 2017.

---

> ### Author Response · Authors · 2018-11-22
> **rebuttal from authors [2/2]**
>
> 4. Regarding out-of-English translation
> We have provided the English-to-many translation results on the IWSLT dataset in the table below. The BLEU scores in () represent the difference between the multilingual model and individual models. Delta represents the improvements of our multi-distillation method over the multi-baseline. We can see our method consistently outperforms the multilingual baseline on all languages, and can nearly match or even surpass the accuracy of the individual models, even if one-to-many translation is considered harder than many-to-one translation. We have also updated the results in the paper.  We will provide more results in the following days.
> ------------------------------------------------------------------------------------------------------------
> Language  |  Individual  |  Multilingual-baseline  |  Multilingual-distill  | Delta
> en-ar          |      13.67       |       12.73 (-0.94)             |       13.80 (0.13) 	      |  1.07
> en-cs          |      17.81       |       17.33 (-0.48)	        |       18.69 (0.88) 	      |  1.37
> en-de         |      26.13       |        25.16 (-0.97)	        |       26.76 (0.63) 	      |  1.60
> en-he         |      24.15       |        22.73 (-1.42)	        |       24.42 (0.27)	      |  1.69
> en-nl          |      30.88       |        29.51 (-1.37)            |       30.52 (-0.36)	      |  1.01
> en-pt          |      37.63       |        35.93 (-1.70)	        |       37.23 (-0.40)	      |  1.30
> en-ro          |      27.23       |        25.68 (-1.55)	        |       27.11 (-0.12)	      |  1.42
> en-ru          |     17.40        |       16.26 (-1.14)	        |       17.42 (0.02) 	      |  1.16
> en-th          |     26.45        |        27.18 (0.73) 	        |       27.62 (1.17) 	      |  0.45
> en-tr           |     12.47        |       11.63 (-0.84)	        |       12.84 (0.37) 	      |  1.21
> en-vi           |     27.88        |       28.04 (0.16) 	        |       28.69 (0.81) 	      |  0.65
> en-zh          |    10.95         |       10.12 (-0.83)	        |       10.41 (-0.54)       |  0.29
> -------------------------------------------------------------------------------------------------------------
>
> 5. Regarding the weight on the distillation loss and turning back the distillation loss
> Thanks for the suggestion. We have provided the results for turning back the distillation loss with a hard threshold in the submitted version. According to the number (Table 7 in the updated version), back distillation improves the accuracy of individual models. We quickly try a simple adaptive weight that changes according to BLEU gap between the teacher and student model: \lambda = 0.9* (1/2)^{max(BLEU_student + 2 - BLEU_teacher, 0)}, which means if a student is lower than the teacher by more than 2 BLEU score, the weight is 0.9. After that, the weight is decayed exponentially. The initial results on IWSLT dataset demonstrate that there is no much difference (with an average of 0.3 BLEU score) compared with the constant weight we used in this paper. We will conduct a comprehensive study on this kind of back distillation you mentioned in the future work.
>
> 6. Regarding the gradient accumulation strategy
> We have conducted experiments to analyze if it is critical for the importance of gradient accumulation. We found it is not critical for the model training. We run an experiment on the IWSLT dataset without gradient accumulation, i.e., updating the model parameters immediately with the training data from a single language. But in order to make sure the update in the two settings has the same batch size, which is a critical hyperparameter for model training, we increase the batch size for the single language by 12 times, to be the same with the batch size in the setting of gradient accumulation. We found the accuracies of the two settings are nearly the same, with 0.3 BLEU higher or lower at most.
>
> 7. Regarding the writing and typos
> We have fixed the typos in the new version. In top-K distillation, the teacher distribution is renormalized.

---

> > ### Comment · AnonReviewer1 · 2018-11-22
> > **okay, good to know**
> >
> > Thanks for running all these extra experiments.

---

> ### Author Response · Authors · 2018-11-22
> **rebuttal from authors [1/2]**
>
> We thank Reviewer 1 for the reviews and comments! Here are our responses to the comments.
>
> 1. Regarding the training time
> The individual models need to be pre-trained, which will incur additional time. According to the training time statistics on IWSLT dataset with NVIDIA V100 GPU, it takes nearly 4 hours to train the individual model with 1 GPU. The total GPU time is 4hours *12 GPUs for 12 languages. The training time for multilingual baseline is nearly 11hours * 4GPUs, while our method is nearly 13 hours*4GPUs. Our method only takes extra 2hours*4GPUs for the multilingual training and 4 hours*12GPUs for the individual model pretraining. Furthermore, we can assume the individual models are pre-given, which is reasonable because the production system usually wants to adapt the already trained individual translation models into multilingual model, at the benefit of saving maintenance cost while with no accuracy degradation or even with accuracy improvement, which is exactly the goal of this work.
>
> 2. Regarding higher-capacity multilingual model
> We have trained larger models on the Ted talk dataset. Our method still consistently outperforms the multilingual baseline model and the individual models, as shown in the table below, where △1 means the BLEU score improvements of our method over the individual models, △2 means the BLEU score improvements of our method over the multi-baseline model.
> ----------------------------------------------------------------------------------------------------------------------------------------
> Language | ar-en | bg-en | cs-en | da-en | de-en | el-en | es-en | et-en | fa-en | fi-en | frca-en
> △1	           | 0.14   | -5.39   | 2.13   | 5.04    | 0.39    | 1.66   |0.69    | 8.46   | 0.32   | 6.78  | 16
> △2               | 1.73   | 3.27    | 0.1     | 1.11    | 1.54    | 0.23   | 0.31   | 0.34   | 1.52  | 0.21   | 0.34
> ----------------------------------------------------------------------------------------------------------------------------------------
> Language | fr-en  | gl-en | he-en | hi-en  | hr-en  | hu-en| hy-en| id-en | it-en | ja-en  | ka-en
> △1	           | 0.37   | 19.38 | -0.17   | 10.49  | 1.64    | 0.35   | 9.59   | 1.58   | 0.19  | 0.14   | 11.59
> △2               | 0.87  | 0.2      | 1.71    | 0.48    | 0.57    |  0.4    | -0.12  | 0.66   | 1.23  | 0.49   | 0.12
> ----------------------------------------------------------------------------------------------------------------------------------------
> Language | ko-en | ku-en | lt-en | mk-en | my-en | nb-en | nl-en | pl-en | ptbr-en | pt-en | ro-en
> △1               | 0.16   | 7.89    | 4.53  | 10.89   | 8.19     | 14.24  | -0.13  | 1.91   | 0.61       | 8.82   | 0.72
> △2               | 1.18   | 0.4      | 0.38  | 0.39     | 0.08     | 0.04     | 0.73   | 0.79   | 0.88       | 1.01   | 0.74
> ----------------------------------------------------------------------------------------------------------------------------------------
> Language | ru-en | sk-en | sl-en | sq-en | sr-en | sv-en | th-en | tr-en | uk-en | vi-en | zh-en
> △1               | 1.29   | 4.58   | 11.98 | 5.16   | 1.79   | 2.46   | 1.37    | -0.04 | 2.13    | 0.25  | 6.96
> △2               | 0.7     | 0.58   | 0.73   | 0.88   | 0.14   | 0.29    | 0.4     | 1.53   | 0.28    | 0.42  | 0.12
> ----------------------------------------------------------------------------------------------------------------------------------------
>
> 3. Regarding corpora size
> There is a typo in Table 9 in the appendix. The training data size for WMT16 En-De is 4.5M bilingual sentence pairs, while for En-Cs is 1M bilingual sentence pairs. We have corrected it in the new version. We used 1M training data for En-Cs in order to make the training data on all languages roughly the same.

---

> > ### Comment · AnonReviewer1 · 2018-11-22
> > **thanks for confirming**
> >
> > Summarizing the responses:
> >
> > 1. So training time expressed in GPU hours does more than double, although wall time can be shorter if enough GPUs are available to train individual languages in parallel. I'm not sure I buy the argument about keeping pre-trained individual models around, since you'd presumably want to re-train if more data or better models become available.
> >
> > 2. Comparing these higher-capacity results to table 3, it looks like the gains over multilingual are smaller. It would be helpful to give average gain, both for table 3 and these new numbers. Also, this isn't very meaningful without knowing what the capacity increase was, and whether you optimized capacity for this setting.
> >
> > 3. If the goal was to have roughly equal corpus sizes, why not use ~5M sentences from En-Cs, rather than 1M?

---

> > > ### Author Response · Authors · 2018-11-23
> > > **reply from authors**
> > >
> > > Thanks for your comments again!
> > >
> > > 1. For the training time, we think it is reasonable to assume the individual models are pre-given. Every production system at least needs to compare the accuracy of the multilingual model with the individual models, to make sure the multilingual model is accuracy enough for online serving. Therefore, the multilingual baseline model also needs the individual models for comparison. In this situation, training the individual models can be not considered into the total training time of our method.
> > >
> > > 2. We just choose the default setting for the transformer model in the original paper [1], without any tuning on the model capacity. For the original model, we use transformer_small with 20M model parameters. For the higher-capacity model, we use transformer_base with 65M model parameters.
> > >
> > > For the improvements, we calculate the average gain over the 44 languages on Ted talk dataset. For transformer_small, △1(distill - individual) is 3.79, △2(distill - baseline) is 1.35. For transformer_base, △1(distill-individual) is 4.25,  △2 (distill-baseline) is 0.68. The rebuttal experiments are actually time limited, so for the higher-capacity model, we just increase the capacity of the multlingual-baseline and multilingual-distil (our method) from transformer_small to transformer_base, keeping the individual models as transformer_small. So the teacher models (individual models) for our method are still transformer_small, and thus the gain over multilingual-baseline (△2) becomes smaller. We can find small teacher models can still teach large student model with improvements, as a byproduct finding of our experiment. We will run the experiments with transformer_base as the individual models (teacher models) to verify that our method can consistently achieve large gain with higher-capacity model.
> > >
> > > 3. For the En-Cs data, there are four data sources for WMT16 En-Cs parallel training data, according to http://www.statmt.org/wmt16/translation-task.html . They are Europarl v7 (647K), Common Crawl corpus (162K), News Commentary v11 (191K) and CzEng 1.6pre (51M). The total training data is 52M. In order to make the training data roughly the same cross different languages, we just choose the first three data, which are 1M in total. We want to make our experiment setting clean and simple, so we do not additionally get part of the data from CzEng 1.6pre (51M) to make it summed to ~5M. We will add this data description in the appendix.
> > >
> > > [1] Vaswani, Ashish, et al. "Attention is all you need." NIPS 2017.

---

> > > > ### Comment · AnonReviewer1 · 2018-11-23
> > > > **Final comment**
> > > >
> > > > I don't have any significant problems with this response, except for the hope that the gains for the current (KD) method over standard multilingual in the high-capacity setting will improve if you also use high-capacity teacher models. The teacher models are trained on much smaller data, so one wouldn't expect them to benefit from higher capacity as much as the multilingual model. Also, it would be interesting to see what happens to the multilingual results if we used an even bigger model like transformer_big. I don't think the results of these experiments should hurt prospects for acceptance, however, because the settings with smaller models and fewer language pairs remain relevant.

---

> > > > > ### Author Response · Authors · 2018-11-26
> > > > > **reply from authors (further results)**
> > > > >
> > > > >
> > > > > We update the results  on the higher-capacity experiment on Ted talk dataset, with transformer_base as the teacher model. The average gain of our distillation method over individual models (△1) is 4.31, and the average gain of our distillation method over multilingual baseline (△2) is 1.06.  You can see as we change the teacher model from small to base, the gains over multilingual baseline also get improved from 0.68 to 1.06. The detailed numbers are shown below.
> > > > >
> > > > > ---------------------------------------------------------------------------------------------------------------------------------
> > > > > Language | ar-en | bg-en | cs-en | da-en | de-en | el-en | es-en | et-en | fa-en | fi-en| frca-en
> > > > > △1	           | 0.05   | -4.42   | 2.09   | 4.91    | 0.16    | 1.76   | 0.92    | 8.53   | 0.56   | 6.95  | 16.2
> > > > > △2               | 2.63   | 4.65    | 0.56   | 1.58    | 1.73    | 0.53   | 0.84    | 0.66   | 1.96   | 0.61  | 0.74
> > > > > ---------------------------------------------------------------------------------------------------------------------------------
> > > > > Language | fr-en  | gl-en | he-en | hi-en  | hr-en  | hu-en| hy-en| id-en | it-en    | ja-en  | ka-en
> > > > > △1	           | 0.71   | 19.37 |  0.2     | 10.37  | 1.59    | 0.86    | 9.32  | 1.51   | 0.22 | 0.33   | 11.62
> > > > > △2               | 1.81   | 0.39   | 2.19    | 0.67    | 0.7      |  1.29   | 0.31  | 1.09   | 1.43 | 0.79   | 0.25
> > > > > ---------------------------------------------------------------------------------------------------------------------------------
> > > > > Language | ko-en | ku-en | lt-en | mk-en | my-en | nb-en | nl-en | pl-en | ptbr-en | pt-en | ro-en
> > > > > △1               | 0.03   | 8.34    | 4.58  | 10.62   | 8.07     | 14.64  | 0.37   | 1.87   | 0.6         | 8.8      | 0.68
> > > > > △2               | 1.25   | 0.8      | 0.47  | 0.5       |  0.35    |  0.54    | 1.33   | 1.09  | 1.22       | 1.39    | 1.14
> > > > > ---------------------------------------------------------------------------------------------------------------------------------
> > > > > Language | ru-en | sk-en | sl-en | sq-en | sr-en | sv-en | th-en | tr-en | uk-en | vi-en | zh-en
> > > > > △1               | 1.28   | 4.55   | 11.58 | 5.02   | 1.69   | 2.19   | 1.59    |  0.03 | 2.05   | 0.03  | 7.08
> > > > > △2               | 0.8     | 0.87   | 0.83   | 1.24   | 0.54   | 0.42    | 0.8     | 1.87   | 0.43  | 0.82  | 0.61
> > > > > ---------------------------------------------------------------------------------------------------------------------------------

---

### Official Review · AnonReviewer3 · 2018-11-02
**Effective knowledge distillation for multilingual NMT, at the cost of increased training time**

**Rating:** 7
**Confidence:** 3

**Review:**

The authors apply knowledge distillation for many-to-one multilingual
neural machine translation, first training separate models for each language
pair. For most language pairs, performance matches or improves upon
single-task baselines.

Strengths:

Improvements upon the baselines are fairly impressive, especially for the
44-language model.

The approach is quite simple and could be easily implemented by other groups.

The paper is well-written and easy to understand.

At inference, only a single model needs to be retained, which is memory-efficient.

Weaknesses:

The authors only test distillation in a many-to-one scenario. I believe that
providing results for many-to-many multilingual NMT would be valuable.

Overall, this approach increases training time as all single-task models
must have converged before beginning distillation.

The authors provide no direct comparison to other work, which makes it hard to
know how strong the baselines are. At least for WMT, I would suggest reporting
results with mteval-v13a (or SACREBLEU), so that results can be compared against
official results.

Questions:

For the top-K approach, do you normalize the top K probabilities so that they
sum to 1 or not?

Did you consider applying sequence knowledge distillation (Kim and Rush, 2016)
(using the baseline beam search output as references) instead of word knowledge
distillation?

***
EDIT: In my opinion, the changes made after the review period clearly improve the quality of the paper. I am increasing my rating from 6 to 7.

---

> ### Author Response · Authors · 2018-11-22
> **rebuttal from authors**
>
> We thank Reviewer 3 for the reviews and comments! Here are our responses to the comments.
>
> 1. Regarding the results of many-to-many translations
> We have provided the English-to-many translation results on the IWSLT dataset in the table below. The BLEU scores in () represent the difference between the multilingual model and individual models. Delta represents the improvements of our multi-distillation method over the multi-baseline. We can see our method consistently outperforms the multilingual baseline on all languages, and can nearly match or even surpass the accuracy of the individual models, even if one-to-many translation is considered harder than many-to-one translation. We have also updated the results in the paper.
>
> We will provide more results on the WMT dataset in the following days and make comparison with previous works.
>
> ------------------------------------------------------------------------------------------------------------
> Language  |  Individual  |  Multilingual-baseline  |  Multilingual-distill  | Delta
> en-ar          |      13.67       |       12.73 (-0.94)             |       13.80 (0.13) 	      |  1.07
> en-cs          |      17.81       |       17.33 (-0.48)	        |       18.69 (0.88) 	      |  1.37
> en-de         |      26.13       |        25.16 (-0.97)	        |       26.76 (0.63) 	      |  1.60
> en-he         |      24.15       |        22.73 (-1.42)	        |       24.42 (0.27)	      |  1.69
> en-nl          |      30.88       |        29.51 (-1.37)            |       30.52 (-0.36)	      |  1.01
> en-pt          |      37.63       |        35.93 (-1.70)	        |       37.23 (-0.40)	      |  1.30
> en-ro          |      27.23       |        25.68 (-1.55)	        |       27.11 (-0.12)	      |  1.42
> en-ru          |     17.40        |       16.26 (-1.14)	        |       17.42 (0.02) 	      |  1.16
> en-th          |     26.45        |        27.18 (0.73) 	        |       27.62 (1.17) 	      |  0.45
> en-tr           |     12.47        |       11.63 (-0.84)	        |       12.84 (0.37) 	      |  1.21
> en-vi           |     27.88        |       28.04 (0.16) 	        |       28.69 (0.81) 	      |  0.65
> en-zh          |    10.95         |       10.12 (-0.83)	        |       10.41 (-0.54)       |  0.29
> -------------------------------------------------------------------------------------------------------------
>
> 2. Regarding training time
> The individual models need to be pre-trained, which will incur additional time. According to the training time statistics on IWSLT dataset with NVIDIA V100 GPU, it takes nearly 4 hours to train the individual model with 1 GPU. The total GPU time is 4hours *12 GPUs for 12 languages. The training time for multilingual baseline is nearly 11hours * 4GPUs, while our method is nearly 13 hours*4GPUs. Our method only takes extra 2hours*4GPUs for the multilingual training and 4 hours*12GPUs for the individual model pre-training. Furthermore, we can assume the individual models are pre-given, which is reasonable because the production system usually wants to adapt the individual translation into multilingual setting, at the benefit of saving maintenance cost while with no accuracy degradation or even with accuracy improvement, which is exactly the goal of this work.
>
> 3. Regarding the top-K distillation
> Yes, we normalize the top K probabilities so that they sum to 1. We have added the description in the new version.
>
> 4. Regarding sequence-level knowledge distillation
> We have tried sequence-level knowledge distillation. It results in consistently inferior accuracy on all languages compared with word-level knowledge distillation used in our work. The results are listed as below.
> -------------------------------------------------------------------------------------------
> Language | Sequence-level distillation | Word-level distillation
> en-ar         |                12.79	                      |             13.80
> en-cs         |                17.01                        |             18.69
> en-de        |                25.89                        |             26.76
> en-he        |                22.92                        |             24.42
> en-nl         |                29.99                        |             30.52
> en-pt         |                36.12	                      |             37.23
> en-ro         |                25.75                        |             27.11
> en-ru         |                16.38                        |             17.42
> en-th         |                 27.52                       |              27.62
> en-tr          |                11.11                        |             12.84
> en-vi          |                28.08                        |             28.69
> en-zh         |                10.25                        |             10.41
> --------------------------------------------------------------------------------------------

---

> > ### Author Response · Authors · 2018-11-26
> > **rebuttal from authors (more results)**
> >
> >
> > We provide the results of one-to-many translation on WMT16 here.  The BLEU scores in () represent the difference between the multilingual model and individual models. Delta represents the improvements of our multi-distillation method over the multi-baseline. We can see our method consistently outperforms the multilingual baseline on all languages, with nearly 1-2 BLEU score gains, and can nearly match the accuracy of the individual models.
> >
> > ------------------------------------------------------------------------------------------------------------
> > Language  |  Individual  |  Multilingual-baseline  |  Multilingual-distill  | Delta
> > en-cs          |      22.58       |        21.39 (-1.19)	        |       23.10 (0.62) 	      |  1.81
> > en-de         |      31.40       |        30.08 (-1.32)	        |       31.42 (0.02) 	      |  1.34
> > en-fi           |      22.08       |        19.52 (-2.56)	        |       21.56 (-0.52)	      |  2.04
> > en-lv          |      14.92       |        14.51 (-0.41)	        |       15.32 (0.40)	      |  0.81
> > en-ro         |       31.67      |        29.88 (-1.79)	        |       31.39 (-0.28)	      |  1.51
> > en-ru         |       24.36      |        22.96 (-1.40)	        |       24.02 (-0.34)       |  1.06
> > -------------------------------------------------------------------------------------------------------------
> >
> > We also compare our baseline with previous works. As we use the transformer_base model, we directly compare our individual baseline with the Transformer paper [1] on En-De translation pair on WMT14 test set.  Our individual model can achieve 27.27 BLEU score while Transformer paper can achieve 27.30, which is comparabe. We also compare our individual baseline with previous work (such as Table 3 in [2]) on WMT16, which are also comparable.
> >
> >
> > [1] Vaswani, Ashish, et al. "Attention is all you need." NIPS 2017.
> > [2] Sennrich, Rico, et al. "The University of Edinburgh's Neural MT Systems for WMT17." arXiv preprint arXiv:1708.00726 (2017).

---

> > > ### Comment · AnonReviewer3 · 2018-11-26
> > > **Clarification needed for BLEU scores**
> > >
> > > I appreciate that you ran one-to-many experiments. I'm a bit confused about the BLEU comparison on WMT. In your paper, you mention using multi-bleu.perl, while [2] uses mteval-v13a.pl, which does its own internal tokenization (on the detokenized input). These two scripts are not equivalent.

---

> > > > ### Author Response · Authors · 2018-11-27
> > > > **reply from authors**
> > > >
> > > >
> > > > Thanks for pointing out. The BLEU scores in [1] are calculated by multi-bleu.perl, so the scores are directly comparable on WMT14 en-de. We have also noticed that mteval-v13a.pl is used in [2]. We have calculated our BLEU scores on WMT16 with mteval-v13a.pl, and found that the BLEU scores by mteval-v13a.pl are just 0.3 BLEU score (on average) less than that calculated by multi-bleu.perl on this dataset. The overall BLEU scores are still comparable (roughly within +/- 0.5 BLEU score).

---

### Public Comment · ~krtin_kumar1 · 2018-12-18
**Question about early stopping**

Based upon the Algorithm 1 ALL loss and NLL loss can oscillate around the threshold point but in Section 3.3 in the discussion about early stopping it seems that you imply that once student improves beyond the threshold then the NLL loss is used always. Which scenario is correct? will there be an oscillation?

---

> ### Author Response · Authors · 2018-12-24
> **reply**
>
> Thanks for your interest in this work.  Yes, once student improves beyond the threshold then the NLL loss is used. But when the accuracy of the student model drops below the threshold, ALL loss is used again. There may be some situation with small oscillation, but not that much, within +/- 0.2 BLEU score. Small accuracy oscillation is common in deep model training. For some cases, when NLL loss is used, the accuracy of student model can be improved again due to multilingual training, away from the oscillation area.

---

> > ### Public Comment · ~krtin_kumar1 · 2018-12-31
> > **reply**
> >
> > Thanks for your reply.
> >
> > So to conclude there will be oscillation with small magnitude and frequency. The reason I am curious about oscillation is because of your results on Ted Talk Dataset,
> >
> > ------------------------------------------------------------------------------------------------------------
> > Language  |    Baseline   |      Early Stopping          |  No Early Stopping
> > Bg               |      27.76       |       29.18 (+1.42)            |       28.07 (0.31)
> > Et                |      14.86       |       15.63 (+0.77)	        |       12.64 (-2.22)
> > Fi                |      16.12       |        17.23	(+1.11)           |       15.13 (-0.99)
> > Fr               |      38.27       |        34.32	(-4.4)              |       33.69 (-4.58)
> > Gl               |      30.32       |        31.9 (+1.58)             |       30.28 (-0.04)
> > Hi               |      19.93       |        21 (+1.07)	               |       18.86 (-1.07)
> > Hy              |      20.25       |        21.17	(+0.92)           |       19.88 (-0.37)
> > Ka              |     16.71        |       18.27 (+1.56)	       |       14.04 (-2.67)
> > Ku              |     11.83        |       13.38 (+1.55)	       |       8.5 (-3.33)
> > Mk             |     31.85        |       32.65 (+0.8)	       |       32.1 (0.25)
> > My             |     13.85        |       15.17 (+1.32)	       |       14.02 (0.17)
> > Sl               |      22.52       |       23.68 (+1.16)	       |       22.1(-0.42)
> > Zh              |     18.81        |       19.39 (+0.58)	       |       17.22 (-1.59)
> > Pl               |      23.5         |       24.3 (+0.8)	               |       25.05 (1.55) (0.75)
> > Sk              |     28.97        |       29.91 (+0.94) 	       |       30.45 (0.94) (0.54)
> > Sv              |    35.92         |       36.92 (+1)	               |       37.88 (1.96) (0.96)
> > -------------------------------------------------------------------------------------------------------------
> >
> > Based upon above results
> > - No early stopping is better in 6/16 cases, out of these the last 3 languages Pl, Sk and Sv have significantly higher scores
> > - For the 3 languages Pl, Sk, Sv no early stopping is better than early stopping, while for other cases early stopping performs better
> > - Thus knowledge distillation seems to be significantly beneficial only 3/16 languages. Since you do not report results on no early stopping on other datasets, I do not know if KD is actually beneficial in other cases.
> > - If KD is not beneficial then the performance improvement that you have achieved is due to early stopping, which in a way tends to pick better of the two (Baseline vs KD), since only using NLL is essentially the baseline model
> >
> > What is your opinion about this conclusion? Do you have results without early stopping on other datasets as well?

---

> > > ### Author Response · Authors · 2019-01-01
> > > **Reply**
> > >
> > > Thanks for your interests in this work.
> > >
> > > However, your conclusion is not correct. Knowledge distillation performs better than the multilingual baseline (purely NLL loss) on all the language pairs you listed.
> > >
> > > Early stopping itself belongs to the distillation method we proposed. Without early stopping can be another alternative of our proposed method, but we have demonstrated  that this alternative will yield worse performance in the paper. So we choose the version with early stopping, which makes sense as you cannot continue to learn from a teacher that is much worse than you (1 BLEU score in this paper) when you gradually improve and surpass your teacher, which will make you worse too.
> > >
> > > After early stopping on a certain language, purely NLL loss is used and some other languages are still using distillation loss. This is not equal to the baseline since the multilingual model now has already achieved better performance than the baseline model due to distillation, and also,  the baseline uses NLL loss on all languages from the beginning while our method switches to NLL loss dynamically depending on the accuracy gap between teacher and student on each language. In this stage, NLL loss is just introduced to ensure the model is not biased to the language still with distillation loss, since if you do not use any loss on the early stopping language, the accuracy will drop due to the model is just trained to optimize other languages.
> > >
> > > At last, you cannot achieve the improvements in this paper with purely NLL loss from the beginning.

---

> > > > ### Public Comment · ~krtin_kumar1 · 2019-01-02
> > > > **reply**
> > > >
> > > > I get your point but let me explain things in a more clear way, there are 3 methods,
> > > >
> > > > Method 1: NLL + KD Loss (without early stopping pure KD)
> > > >
> > > > Method 2: NLL+KD or only NLL (early stopping or oscillation with small mag. and freq.) (this is not pure KD, sometimes and for some languages NLL loss (baseline) is used)
> > > >
> > > > Method 3: NLL loss only (Multilingual)
> > > >
> > > > We know on Tedtalk dataset Method 2 > Method 3 > Method 1
> > > >
> > > > This might indicate that Method 2 i.e. early stopping is the cause of significant improvement and KD might not be helping a lot since Method 1 > Method 3 and method 1 uses KD loss always.

---

> > > > > ### Author Response · Authors · 2019-01-05
> > > > > **reply**
> > > > >
> > > > > Method 2 is exactly what we proposed. Our work is not a simple application of KD for multilingual NMT (e.g., method 1). Instead, we need to carefully design the distillation method, including when to distill and which language to distill.

---

### Meta-Review · Area_Chair1 · 2018-12-14
**Accept**

**Confidence:** 4
**Recommendation:** Accept (Poster)

**Metareview:**

This paper presents good empirical results on an important and interesting task (translation between several language pairs with a single model). There was solid communication between the authors and the reviewers leading to an improved updated version and consensus among the reviewers about the merits of the paper.